# PHERI—Phage Host ExploRation Pipeline

**DOI:** 10.3390/microorganisms11061398

**Published:** 2023-05-26

**Authors:** Andrej Baláž, Michal Kajsik, Jaroslav Budiš, Tomáš Szemes, Ján Turňa

**Affiliations:** 1Geneton Ltd., Ilkovicova 8, 841 04 Bratislava, Slovakia; andrej.balaz@fmph.uniba.sk (A.B.); jaroslav.budis@geneton.sk (J.B.); tomas.szemes@geneton.sk (T.S.); 2Department of Applied Informatics, Faculty of Mathematics, Physics and Informatics, Comenius University, Mlynska dolina F1, 842 48 Bratislava, Slovakia; 3Science Park, Comenius University, Ilkovicova 8, 841 04 Bratislava, Slovakia; jan.turna@uniba.sk; 4Department of Molecular Biology, Faculty of Natural Sciences, Comenius University, Ilkovicova 6, 841 04 Bratislava, Slovakia; 5Medirex Group Academy n.o., Novozamocka 1, 949 05 Nitra, Slovakia; 6Slovak Centre of Scientific and Technical Information (SCSTI), Lamacska Cesta 8/A, 811 04 Bratislava, Slovakia

**Keywords:** bacteriophages, machine learning algorithm, phage host determination

## Abstract

Antibiotic resistance is becoming a common problem in medicine, food, and industry, with multidrug-resistant bacterial strains occurring in all regions. One of the possible future solutions is the use of bacteriophages. Phages are the most abundant form of life in the biosphere, so we can highly likely purify a specific phage against each target bacterium. The identification and consistent characterization of individual phages was a common form of phage work and included determining bacteriophages’ host-specificity. With the advent of new modern sequencing methods, there was a problem with the detailed characterization of phages in the environment identified by metagenome analysis. The solution to this problem may be to use a bioinformatic approach in the form of prediction software capable of determining a bacterial host based on the phage whole-genome sequence. The result of our research is the machine learning algorithm-based tool called PHERI. PHERI predicts the suitable bacterial host genus for the purification of individual viruses from different samples. In addition, it can identify and highlight protein sequences that are important for host selection.

## 1. Introduction

Bacterial infections have affected public health throughout human history. The introduction of antibiotics reduced human morbidity and mortality caused by infectious diseases dramatically. However, the emergence of multidrug-resistant pathogenic bacteria reverted the situation once again. Moreover, the problem of multi-drug resistance is getting worse. WHO calls attention to infections, especially by *Klebsiella pneumoniae*, *Mycobacterium tuberculosis*, and *Neisseria gonorrhoeae*, and blood poisoning and foodborne diseases. These infections are becoming harder and sometimes nearly impossible to treat [1]. Moreover, antibiotic resistance is now recorded in every country [2]. One of the possible solutions is the use of bacteriophages in therapy. Phages have relatively simple structures composed of proteins (approx. 60%) that encapsulate a DNA or RNA genome (40%) [3,4]. Phages are among the most abundant entities in the biosphere, with an estimated 10^31^–10^32^ phages in the world at any given time. Moreover, they play a crucial role in regulating bacterial populations; for example, phages are responsible for the death of approximately 20–40% of all marine surface bacteria every 24 h [5,6,7]. They are ubiquitously and naturally distributed in all environments populated by bacterial hosts, including soil, water, air, and the intestines of humans and other animals [7,8,9,10]. The idea of using bacteriophages in therapy is not new. Phage therapy has been used in the former Soviet Union countries for decades [11,12], but in the last few years, it has begun to be applied in Western countries as well. Bacteriophages have proved their usefulness not only in animal models such as mice [13,14], cattle [15,16], chicken [17], zebrafish [18], and dog [19], but also when used in humans. Human phage therapy has gained reliance through research projects such as PhagoBurn [20] or practical experience in Georgia [21], leading to the first cases of phage use on patients in Western countries. In recent years, phage therapy has been successfully used for the intravenous treatment of bacterial infections in cystic fibrosis patients in the US and Georgia. It has been used against multi-drug resistant pathogens such as *Achromobacter xylosoxidans*, *Pseudomonas aeruginosa*, *Mycobacterium abscessus*, and *Burkholderia dolosa* [22,23,24,25]. Additionally, in the treatment of *Mycobacterium abscessus*, the patient was treated with a cocktail of three phages, of which only one was naturally lytic, and the other two were engineered to increase their lysis efficiency by deleting the receptor gene or its HTH domain [25]. These studies used well-characterized phages from collections, with known hosts, one of the primary conditions for their successful practical use. Characterized phages were amplified on the host, purified, and subsequently sequenced. However, the introduction of high-throughput sequencing has allowed us to examine metagenomic colonies of bacteria right from the environment. This method has the potential to discover a considerable number of new species, which were not cultivable before. However, it also produces more and more phage genomic sequence data without an identified host. Luckily, the host range of known phages tends to be relatively narrow, often consisting of only a subset of strains making up a single bacterial species [26]. The problem of unknown hosts can be alleviated by a bioinformatic approach. For a successful phage infection, it needs not only to adsorb to the host surface and insert its genetic information, but it also needs to overcome its immune response and ensure successful transcription and translation. It is therefore essential to consider changes in bacterial surface structures and thus in the presence of phage receptors on individual strain membranes within the species [27]. Equally important is the consideration of the bacterial host immune response in terms of restriction-modification systems [28,29], the CRISPR mechanism [30] or abortive systems [31,32]. Additionally, factors affecting phage gene transcription and translation are important, such as the availability of specific tRNA or sufficient amino acids. However, the change in specificity may be due to overcoming the host response, as in obtaining the resistance of the CRISPR system [33]. All these parameters can negatively affect the host prediction. Nevertheless, several groups have already attempted to bioinformatically elucidate the phage-host interaction using a variety of approaches and tools such as Virsorter [34], MGTAXA [34,35] or HostPhinder [36]. Our goal was also to create a bioinformatic tool for predicting the host from the whole genome sequence, but we chose the machine-learning-algorithm approach. The use of machine learning algorithms has proved to be suitable for phage biology, as evidenced by their use in the search for phage virions [37], and improved phage genome annotation [38] as well as phage classification [39,40]. Our pipeline, PHERI, re-annotates phage genomes, and uses TRIBE-MCL for rapid and accurate clustering of annotated protein sequences [41,42] and a binary decision tree classifier to predict the phage host. The rationale behind our method lies in the close relationship between the genomic sequence of a gene and the biological function of its translated protein. Even if the function of the gene is unknown, the presence of similar sequences in phages infecting the same hosts indicates that the mentioned sequences are related to the host specificity. The presence of such sequences in the tested genome may resolve potential host infiltration.

## 2. Materials and Methods

We used phage genome sequences collected from several publicly available databases to identify clusters of gene sequences that are specific for distinct hosts. We used them to classify unlabeled phage genomes using the following analysis steps. At first, coding regions of the genomic sequence were identified and extracted. Then, they were compared with gene sequences collected from phages with known hosts, resulting in the reduced binary vector representation of the phage. Finally, the vector was classified to identify potential hosts of the phage.

### 2.1. Collection of Phage Sequences

We downloaded genomic sequences of phages from three publicly available databases using automated in-house scripts; 6091 records from GenBank [43], 2070 records from ViralZone [44,45] and 2567 records from PhageDB [46]. Although these databases cover the majority of currently sequenced and published phages, we made the downloading step easily extensible for adding new sources of phage sequences that may emerge in the future. Downloaded records were highly redundant, mainly because many phage sequences were simultaneously presented in more databases. Therefore, we merged downloaded datasets and removed duplicated records, resulting in a non-redundant dataset of 7064 phage sequences capable of infecting 183 bacterial genera (Figure 1). The host name and taxonomy for each sequence were obtained and unified according to NCBI taxonomy to allow computer processing. The phages with hosts from the 50 most abundant bacteria genera were selected for further analysis. The phages outside this group were discarded due to an insufficient number of samples for machine learning analysis. Genomes were further divided, using random sampling, into two distinct datasets; training set with 4723 (80%) sequences and testing set with 1202 (20%) sequences. The training set was utilized to identify clusters of common gene sequences and train parameters of the classifier. The accuracy of the method had been validated on the testing set (Table 1).

### 2.2. Extraction and Annotation of Genes

Phage genome sequences were annotated with locations of genes and their biological functions. Although gene annotations of particular genomes are part of genomic records in the utilized databases, we decided to annotate sequences from scratch. This way, we ensured consistency of annotations across our data with up-to-date knowledge. We used a publicly available pipeline called Prokka [47] to identify and annotate genes. First, coordinates of coding DNA sequences (CDS) were found with the Prodigal tool [48]. After the locations of genes were predicted, Prokka can start to annotate functions of all CDSs. The tool does this by comparison of a sequence to several databases of proteins with an experimentally determined function [45,49] or pre-processed protein families and domains [45,50,51]. The Prokka pipeline was run with the parameter—kingdom Viruses to consider that we were annotating genes in phages.

### 2.3. Clustering of Gene Sequences

We compared extracted genes to identify clusters of highly similar sequences, presumably with the same biological function. Since thorough pairwise comparison of all proteins in the training set would be overly time-consuming, we employed a multi-step approach. In the first step, genes were deduplicated using CD-hit [52] to reduce the enormous number of sequences. Then, gene pairs with at least some local sequence similarity were identified using an optimized implementation of the Blast alignment tool [42], called CrocoBLAST [53] (used parameters blastp -max_target_seqs 1,000,000 -max_hsps 1. All other parameters are default). Afterwards, the identified pairs with high local sequence similarity underwent thorough pairwise alignment [54] to retrieve more accurate similarity scores. Based on these similarity scores, we identified gene clusters using the Markov Cluster Algorithm implemented in the MCL package [41]. We recovered 32,281 gene clusters. A substantial portion of clusters contained only a small number of gene sequences. We think such clusters do not have enough information to significantly help the classifier and only increase the chance of overfitting. Therefore, we removed all clusters with genes from less than 1% of phages from the training set. In the result, we obtained 1965 clusters that were used further in the classification.

### 2.4. Training Classification Model

We trained a binary decision tree classifier [55] for each bacterial host from the dataset separately, since a united classifier for all potential hosts would be too complicated for coherent interpretation. Additionally, the phage sequence may be labelled with multiple bacterial hosts. The separate classification allows one to label less-specific phages with multiple admissible hosts. At first, phage sequences from the training set were transformed to the reduced integer vector representation, where value a_i,j_ represents a number of genes from cluster j belonging to phage i. For each host, we trained a classifier to predict if an input vector represents a phage that can infect a given host. Each node in the resulting decision tree represents a single gene cluster with informative value regarding phage specificity. The presence or absence of such genes guides decision along the tree. Each informative cluster may be annotated with biological function to improve the interpretation of the decision process. Gene clusters without known function are good candidates for follow-up experimental evaluation. The tool PHERI, the source code for the model training as well as the source code for the tool is available at https://github.com/fmfi-compbio/pheri (accessed on 10 May 2023).

### 2.5. Classifying Novel Sequence

A novel phage genome sequence is classified using similar steps. At first, gene sequences are identified using Prodigal tool. Then, collected gene sequences are compared with the gene clusters using Blast. Genes with significant matches with any sequence from a cluster are assumed to be members of clusters. Finally, the set of genes is transformed into a reduced vector representation. The vector is labelled with all trained classifiers. All bacteria with a positive classification are assumed as potential hosts for the phage.

### 2.6. Bacterial Strains and Growth Conditions

*Escherichia*, *Cronobacter*, *Enterobacter*, *Salmonella*, *Klebsiella*, *Staphylococcus*, *Proteus*, *Morganella* and *Citrobacter* strains were isolated from clinical or food samples in our laboratory or were obtained from the collections of Nottingham Trent University, UK, the Belgian Coordinated Collections of Microorganisms, the Czech Collection of Microorganisms or from Slovak Food Research Institute. We used a total of 33 strains of the genus *Escherichia*, 26 *Cronobacter*, 15 *Salmonella*, 5 *Enterobacter*, 3 *Proteus*, 3 *Staphylococcus*, 2 *Klebsiella*, 2 *Pseudomonas* and 1 *Morganella morganii* strain. We later expanded the set of strains by five bacteria from the genus *Citrobacter*. The Luria-Bertani (LB) broth and LB agar were the general-purpose media used to cultivate strains.

### 2.7. Isolation of Bacteriophages

Bacteriophage DevCS701 (ON157416) [56] was isolated from a sample from the Bratislava wastewater treatment plant on the indicators *C. sakazakii* NTU701. Wastewater was sterilized by passage through a 22 μm filter and mixed with an equal volume of twofold-concentrated LB medium and up to 1% of overnight bacterial culture. The inoculated mixture was cultivated overnight at 37 °C with shaking. Phages were purified by three repeated isolations from single plaques on double agar, followed by ultracentrifugation in a CsCl gradient [57].

### 2.8. Plaque Assay and Host Range

The 200 μL overnight bacterial culture was supplemented with 10 μL of 1 M CaCl_2_ and 10 μL of 1 M MgCl_2_, mixed with 5 mL of top agar (0.2% peptone, 0.7% NaCl and 0.7% agar), and poured onto an LB agar plate. A volume of 10 μL of the appropriate bacteriophage suspension (102–1010 PFU/mL) was spotted onto the plate and incubated overnight. Alternatively, 20 μL of bacteriophage suspension was mixed with 200 μL of overnight bacterial culture and with 5 mL of top agar and poured onto the LB agar. After overnight cultivation at 37 °C, the plaques were counted. The strain *C. sakazakii* NTU701 was used as a reference for the determination of the efficiency of plating (EOP) for phage Dev-CS701.

### 2.9. Phage Adsorption

A volume of 180 μl overnight bacterial culture (OD600 = 1) was mixed with 20 μL of phage suspension (108 PFU/mL; the multiplicity of infection = 0.001) at 37 °C. After 10 min, 10μL of the sample was diluted in 0.99 mL of cold SM buffer (100 mM NaCl, 8 mM MgSO_4_, 50 mM Tris-HCl, pH 7.5, 0.002% gelatine) and centrifuged. Unabsorbed phages from supernatants were counted by plaque assay, and the amount of phage adsorbed was calculated as the percentage of cell-bound phage. The measurements were repeated in triplicate.

## 3. Results

In this study, we developed and benchmarked PHERI, a tool for predicting the bacterial host genera for phages from metagenomic data. The method is based on the assumption that phages infecting the same host share similar protein sequences. In making predictions, PHERI relies on a reference database, in which sequences of phages have been annotated, and resulting genes have been categorized based on similarity. Subsequently, host-specific clusters and cluster combinations were searched and 50 genus-specific decision trees were created. The genomes of the tested phages, for which the hosts should be predicted, were also annotated, categorized and compared to decision trees. If the test bacteriophage contained sequences similar to those in clusters in a specific tree, we assumed that it could infect the host for which the decision tree is created. The prediction was then experimentally verified on phage samples from our collection.

### 3.1. Developing PHERI Method

The method uses a reference database that we made of unique phage sequences from three databases—GenBank, ViralZone and PhagesDB. In the exploratory stage of our analysis, we used python library scikit-learn [58] for principal component analysis. Reduced representation of phages in the form of an integer matrix was used as an input. The first few principal components were used to create plots in the python library matplotlib. In Figure 2, we can see data visualized with the first principal component on the x-axis and the second principal component on the y-axis. Each data point represents one phage record, and the colour of the particular data point corresponds to the genus of that phage. Most phage records are located around the centre, with some distinct groups of Mycobacterium and Staphylococcus phages outside the centre. The first two principal components retained less than 21% of the dataset variability (Figure 2). This result suggests that many predictors are contributing valuable information to the dataset. To take into account this high number of predictors and at the same time keep the interpretability of the resulting model acceptable, we chose Decision Tree Classifier as our model.

Before the feature selection, our training dataset consisted of a matrix with 4723 rows, representing phages, and 32,281 columns, representing gene clusters. The high dimensionality of our data could lead to the increased probability of the overfitting of models on data. To address this concern, we decided to perform feature selection as a process of removing dimensions with low importance from the dataset. The reason to prefer feature selection over feature extraction methods as PCA presented in the previous section was that we wanted our tree models to be interpretable in terms of important clusters rather than in terms of principal components. Because we expected many clusters with a small number of genes, our choice for the feature selection method was the Variance Threshold. The Variance Threshold method is a simple method that removes columns with variance under a certain threshold. Using this technique, we removed columns in the matrix with ones in more than 99% of cases or with zeros in more than 99% of cases. The reduced matrix had 4723 rows and 1965 columns. In our work, we used a Decision Tree Classifier implementation from the python library scikit-learn. For each group of phages with hosts from selected genera, we created one model. Each of those models was trained to answer a question, whether one particular phage was able to infect bacteria from a specific genus. Models were trained with a reduced matrix used as the feature. The ability to infect a specific genus was used as the label. To prevent overfitting of our trees, we also set the parameter min_impurity_split to the value 0.03. This parameter enabled a threshold for splitting leaves and therefore only nodes with an impurity index greater than 0.03 were divided. The threshold 0.03 was determined empirically. Lower values created a tree with many nodes, where the risk of overfitting was high and greater values did not have enough nodes to maintain a model’s accuracy. With this approach, we created a model for each of our 50 selected host genera and visualized it with the python library Graphviz. For classification, we expected to have a complete sequence of bacteriophage.

### 3.2. Host Prediction Evaluation

To examine the accuracy of our models, we classified all bacteriophages from our test dataset. Test dataset contained 1202 phage records. Resulting predictions were aggregated and the number correctly predicted (TP + TN), false-positive (FP), false-negative (FN), sensitivity (TPR = TP/P), specificity (TNR = TN/N) and informedness (BM = TPR + TNR − 1) was recorded. From the identified values, we determined the accuracy, sensitivity, specificity and informedness prediction for 50 bacterial genera with the highest number of infecting phages. PHERI predicted a host of bacteriophages infecting Leuconostoc, Reugenia and Helicobacter the best. Accuracy, sensitivity and specificity were equal to or close to 100%. At the opposite end of the spectrum were bacteriophages infecting the genera Stenotrophomonas and Citrobacter (Table 1, Figure 3).

### 3.3. Comparison of PHERI to Other Tools

Surprisingly, there are not many tools with the same goal as PHERI. We discuss some notable examples in the Discussion. Here, we provide a comparison with the tool HostPhinder, where the goal is the most aligned with ours. We calculated the accuracy in the testing dataset, which was held out at the beginning of the analysis. This hold-out testing dataset mimics the conditions during its use in practice in the laboratory, where the only information we have is the sequence of the isolated phage. The accuracy of both tools was remarkably high (>0.85). In Figure 4, we show the accuracy across all 50 selected genera, with the accuracy of HostPHinder on the x-axis, the accuracy of PHERI on the y-axis and the genera marked as blue points. PHERI achieved consistently high accuracy around 0.98 even for more challenging targets, where HostPhinder’s accuracy dropped below 0.90 (Figure 4).

### 3.4. Host Prediction for New Isolated Bacteriophages

The PHERI’s functionality was also verified by determining the host of phages isolated in our lab and not added to the public databases. Tested bacteriophages were isolated from wastewater from Bratislava, Slovakia. Their host specificity, as well as whole-genome sequence, was previously determined using standard wet science methods. The bacterial host genus for five out of six phages was successfully predicted using the PHERI method. Moreover, for the DevCS-701 phage, PHERI determined different bacterial genera (Table 2).

According to laboratory tests, bacteriophage Dev-CS701 infects isolates from the genus Cronobacter, although PHERI predicted isolates from Citrobacter genus as the most likely candidate. For this reason, the host panel was expanded to include Citrobacter strains and re-established specificity. The extended host panel proved PHERI prediction since the Dev-CS701 phage infected a representative of the genus Citrobacter, namely Citrobacter gillenii CCM 4711. However, the bacteriophage was not able to infect all Citrobacter strains. For this reason, we also examined the bacteriophage adsorption rate to the tested isolates (Figure 5). Dev-CS701 was able to bind to six out of seven Cronobacter strains and two out of four Citrobacter strains as well. Other strains did not reach high values, but the increased rate of adsorption on Enterobacter strains is also interesting.

Despite partial proof of the accuracy of the prediction, we decided to determine the cause of the selection of the genus Citrobacter instead of the genus Cronobacter by examining the decision trees, see Appendix A. The phages contain sequences classified into clusters from both Citrobacter and Cronobacter decision trees; in fact, clusters 54 and 170 were found in both trees, see Appendix A. In total, the phage had six sequences that were classified into five clusters for both decision trees. However, PHERI was able to classify phages only according to the Citrobacter decision tree.

## 4. Discussion

The study of bacteriophages could help solve many of the problems with resistant bacteria in medicine, veterinary, food and other industries. One of the basic criteria for bacteriophage utilization is the knowledge of their whole genome sequence as well as the host [59,60]. The classic bacteriophage characterization methods were based on phage studies with a known host range and subsequent sequencing. However, since the advent of new massively parallel sequencing methods, these procedures are often reversed. It is also possible to identify bacteriophages that infect non-cultivable bacteria, the so-called “bacterial dark matter”. Our studies have been previously focused on the identification of specific bacteriophages capable of infecting foodborne pathogens [61,62]. However, by using metagenomic data in our research, we discovered phages without the known host. Therefore, we developed a bioinformatic pipeline for predicting the phage bacterial host genus from the whole-genome sequences, PHERI, based on machine learning algorithms. A couple of groups have already tested the idea of using a bioinformatic approach to identify phage hosts. For example, Martínez-Garcia et al. described one possibility of identifying a host without cultivation. They retrieved genomic content of individual cells from an environmental sample using single-cell genomic technologies, then hybridized them against a set of phage genomes from the same sample, immobilized on a microarray and sequenced positive hybridization cases. Using this method, they were able to pinpoint viruses infecting the ubiquitous hyper-halophilic *Nanohaloarchaeota*, included in the so-called ‘microbial dark matter’ [63]. Another approach of the virus–host adaptation analysis was chosen by Roux et al. They developed a bioinformatics tool for virus sequence identification. The VirSorter identified prophage sequences through a combination of the detection of hallmark viral genes, enrichment of viral-like genes, depletion in PFAM-affiliated genes, enrichment in uncharacterized genes, enrichment in short genes and the depletion in strand switching [34]. This tool was able to identify 12,498 virus–host linkages from almost 15,000 bacterial and archeal genomes. Identified prophage sequences came from 5492 microbial genomes, and provided first viral sequences for 13 new bacterial phyla. In their study, they also analysed the virus–host adaptation in compositions in terms of nucleotide frequency and codon usage showing the strongest signal of adaptation to the host genome given by tetranucleotide frequency (TNF) [64]. Another classification method to predict the taxonomy of bacterial hosts for uncharacterized viral metagenomic sequences, that does not rely on homology or sequence alignment, was developed by Williamson et al. In their study explaining the composition of the marine virome in the Indian Ocean, they also described the bioinformatic tool MGTAXA, which links phage sequences to the highest-scoring bacterial taxonomic model based on polynucleotide genome composition similarity between the virus and host genomes [35]. An excellent tool for host prediction is also HostPhinder created by Villarroel et al. The HostPhinder is based on the assumption that genetically similar phages are likely to share bacterial hosts. The tool utilizes a phage database with known sequences that are divided into k-mers. Phages with an unknown host are also divided into k-mers and compared to a database. The high similarity of short DNA sequences between two phages will determine the likely host [36]. PHERI bases its predictions on the machine learning technique, Decision Tree Classifier. The advantages of this technique are its interpretability and the potential for rapid improvement in prediction accuracy in the era of big datasets. The dependence of this technique on the size and quality of the dataset may also be considered its biggest drawback, as we showed for genera with small amounts of data. In this work, we used phage whole-genomics records from public databases with known hosts to create clusters of similar gene sequences that are specific to a certain genus. Sequences were annotated using the Prokka tool [47], and genes were extracted using a custom script. Extracted genes were aligned using BLAST with a database of genes from the training set. Based on the retrieved similarity, clusters of genes were created. After that, a vector of integers was created for each phage representing genes and their corresponding assignment to clusters. This vector was passed to the decision tree model, and the resulting prediction was saved. For each of the 50 genera tested, a decision tree based on the necessity of specific clusters to infect the genus was created. Considering the mosaic structure of phage genomes, one of the advantages of using machine learning algorithms for phage host predictions is that only presence, absence and quantity of genetic elements influence the outcome. Thus, differences in locating and organizing individual genes do not affect the outcome of the pipeline prediction. In the evaluation test consisting of 1202 phages from the database, PHERI performed well when it reached the accuracy of 99.37% for the host genus prediction. However, the differences between bacterial genera were considerable as some hosts were easier to predict than others. We noticed a more accurate prediction of host genera with more than 200 phages in the database. The average sensitivity of prediction here was 81.8%. The prediction was less sensitive for families with more than 100 and more than 10 phages, reaching 74.4 and 50.7%, respectively. The data show that more representatives in the database increase the accuracy of the prediction. This result is probably due to the greater number of different host-specific protein sequences that PHERI clustered and incorporated into the decision tree. A greater number of clusters in the tree reduces the likelihood of incorrect prediction in the case of a phage with different mechanisms of infection. Chibani et al. has already described improvements in the prediction of machine learning algorithms based on the number of phages in the database in their phage classification study [39]. The small number of phages that infect individual species was the main reason why we designed PHERI to identify genera. In this way, we were able to increase the accuracy of the prediction and thus allow narrowing the range of hosts for later wet-science host specificity tests. At the same time, we assume that by increasing the number of specific phages in the database, PHERI has the potential not only to increase the accuracy of genus prediction but also to predict the host at the species level. The number of specific phages in the database was not the only factor affecting the accuracy of the prediction. In particular, PHERI has identified all phages infecting the genus *Leuconostoc*, which has only 17 specific phages in the database. In contrast, in the case of the genus *Stenotrophomonas* with 13 phages in the database, none could be identified. By comparing bacteriophages infecting *Leuconoctoc*, we found that leuconostoc-specific phages form two highly related groups belonging to the genera *Limdunavirus* and *Unaquatrovirus* within the subfamily *Mccleskeyvirinae.* Homologous phages probably have a similar mechanism of infection that provides similar proteins. A similar conclusion was reached by Kot et al. in the comparative genomic analysis of *Leuconostoc* phages [65]. PHERI, therefore, constructs a decision tree for a group of genetically related phages easier and does not need a large number of viruses in the database. Similar results were obtained with the prediction of phage hosts of other bacterial genera with a small number of specific viruses. For example, the genera *Paenibacillus* or *Ruegenia* with 26 and 12 genetically related viruses achieved a sensitivity of prediction over 99%. By contrast, phages infecting *Stenotrophomonas* are not genetically related, since some, such as IME13 (NC_029000.1) phages, belong to the *Straboviridae* family [66], phage vB_SmaS_DLP_5 (NC_042082.1) belongs to the *Delepquintavirus* genus within *Caudoviricetes* class, or phages such as PSH1 (NC_010429.1) belong to the *Inoviridae* family [67]. Narrow and variable groups of phages in the training set does not allow one to construct a reliable decision tree. Another factor that could affect the accuracy of the prediction is the ability of bacteriophages to infect bacteria of different species, even genera. Especially in cases of genetically related bacterial genera, several instances of phages with the ability to infect multiple genera have been described [62,68,69]. Even in these cases of known cross-genera host specificity, only one genus name was found in the database. We used PHERI to locate a host of several phages isolated and characterized in our laboratory. The phages had established host specificity for strains of the genera *Escherichia*, *Cronobacter*, *Enterobacter*, *Salmonella*, *Klebsiella*, *Staphylococcus*, *Proteus*, and *Morganella*. Pheri correctly identified the host genus for five out of six phages (Table 2). In the case of the phage Dev-CS701, which infected strains of the genus *Cronobacter*, it predicted as a suitable host the bacteria from genus *Citrobacter*. Subsequent extended host specificity tests against strains of the genus *Citrobacter* confirmed that the phage also infected *Citrobacter gillenii* CCM 4711. Unfortunately, the phage was unable to form plaques on other strains of the genus. We therefore tested the bacteriophage’s ability to recognize the bacterial surface, which confirmed that Dev_CS701 recognizes the surface not only of *C. gillenii* but also of *C. werkmanii*. In addition, a comparison of whole-genome sequences of phages by BLAST showed that Dev_CS701, in addition to its closest relative cronophage vB_CsaM_IeB (KX431559.1) [70], achieved similarity of over 96% to the Citrobacter-specific phages Margaery (KT381880.1) and Maroon (MH823906.1) [71]. Unfortunately, the detailed host specificity of the closest-related phages is not yet publicly available. With our tool, we wanted to show a new method in the prediction of phage hosts. PHERI can help isolate live viruses from samples in wet labs by narrowing the range of possible hosts. “There is also the potential to refine the prediction with the increasing number of new phages in databases. With the help of more accurate data from the database and the addition of several bacterial genera and species, we plan to increase the accuracy of the tool to the level of bacterial species. One of the advantages of host prediction based on the clustering of individual genes is the possibility of highlighting genes with unknown functions necessary for infection. The identification of such genes may, in the future, help scientists elucidate the mechanisms of infection by individual bacteriophages. Moreover, by identifying these genes, it will be possible to study them better or directly use them for the preparation of recombinant phages with changed properties”.

## 5. Conclusions

The importance of bacteriophages as a research subject is rising mainly due to the decreasing effectiveness of antibiotic treatments. Bacteriophages could be used as novel weapons in the fight against bacterial infection. The goal of this work was to examine relationships between bacteriophage genomes and their bacterial hosts. The tool created is proof of the concept that the machine learning algorithm-based tool can be used to search for bacterial hosts for viruses. PHERI can accelerate the isolation of individual bacteriophages from samples by narrowing the range of potential hosts. It also provides the user with information about likely proteins that are required for a successful infection.

## Figures and Tables

**Figure 1 microorganisms-11-01398-f001:**
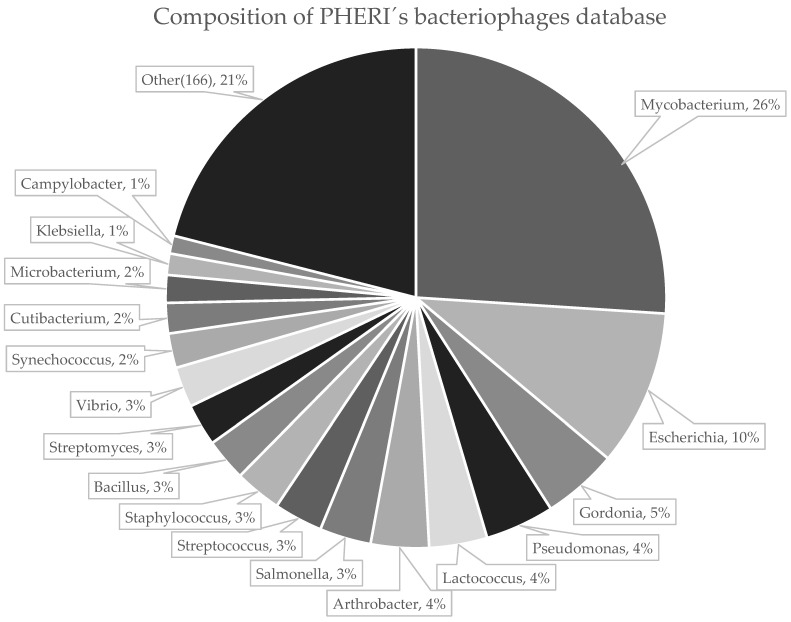
Composition of hosts infected by bacteriophages from the PHERI database. The database is made up of bacteriophages infecting at least one representative of 183 bacterial families.

**Figure 2 microorganisms-11-01398-f002:**
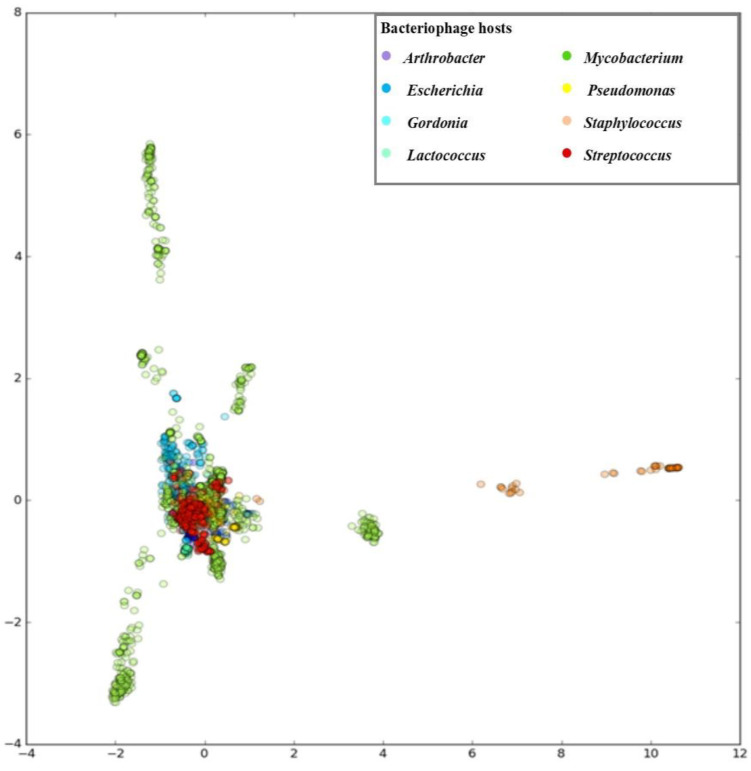
Principal component analysis. First two principal components, PC1 (11.57% of variability) on the *x*-axis, and PC2 (9.19% of variability) on the *y*-axis.

**Figure 3 microorganisms-11-01398-f003:**
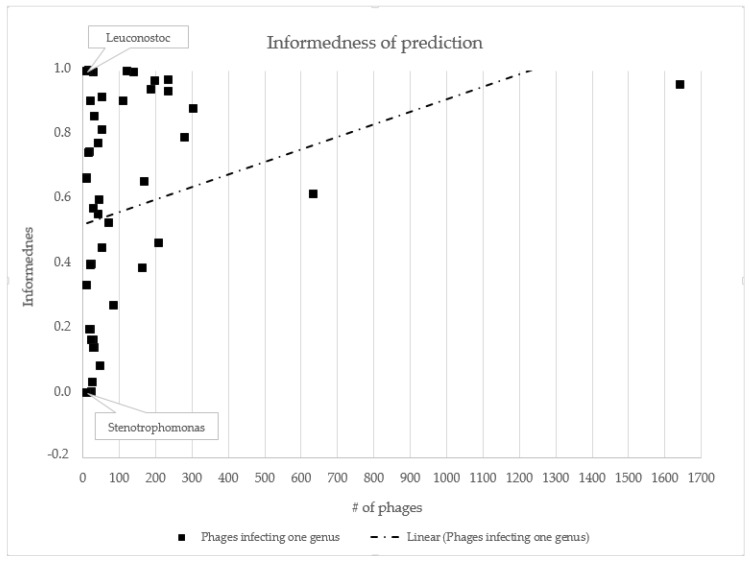
Informedness of PHERI host prediction. Each point represents phages infecting one bacterial genus. Value informedness estimates the probability of an informed decision; the closer the values are to one, the more credible the prediction is. The trendline confirms the hypothesis that the informedness value increases with the growing number of specific phages.

**Figure 4 microorganisms-11-01398-f004:**
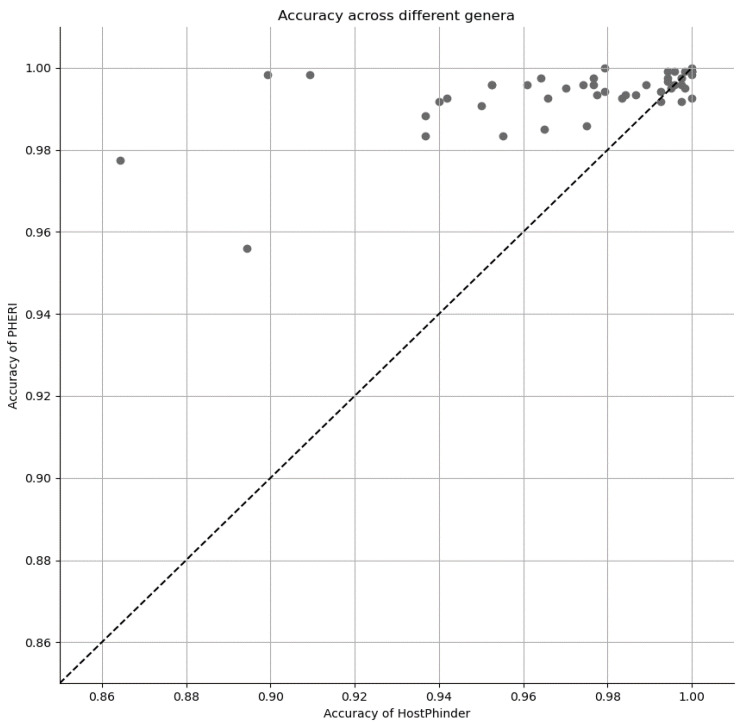
Accuracy of host prediction across 50 bacterial genera.

**Figure 5 microorganisms-11-01398-f005:**
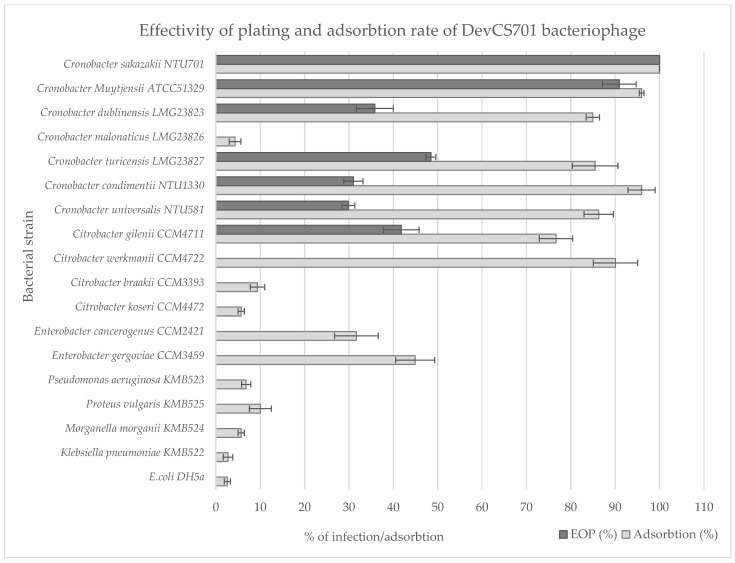
Effectivity of plating and adsorption rate of DevCS701 phage on various hosts.

**Table 1 microorganisms-11-01398-t001:** Number of true positive, true negative, false positive and false negative predictions on the testing dataset (*n* = 1202).

Genus	True Positive	True Negative	False Positive	False Negative
*Leuconostoc*	4	1198	0	0
*Ruegeria*	3	1197	2	0
*Helicobacter*	6	1194	2	0
*Paenibacillus*	6	1192	4	0
*Cutibacterium*	25	1173	4	0
*Moraxella*	7	1190	5	0
*Synechococcus*	29	1166	7	0
*Lactococcus*	47	1145	9	1
*Streptococcus*	39	1152	10	1
*Mycolicibacterium*	320	847	33	2
*Staphylococcus*	37	1155	8	2
*Arthrobacter*	45	1150	4	3
*Rhodococcus*	11	1189	1	1
*Microbacterium*	21	1171	8	2
*Bacillus*	32	1156	11	3
*Gordonia*	55	1135	5	7
*Flavobacterium*	6	1194	1	1
*Acinetobacter*	9	1188	3	2
*Pseudomonas*	46	1129	16	11
*Aeromonas*	7	1190	3	2
*Corynebacterium*	3	1194	4	1
*Caulobacter*	3	1193	5	1
*Proteus*	2	1199	0	1
*Mannheimia*	2	1197	2	1
*Streptomyces*	23	1164	3	12
*Escherichia*	82	1044	31	45
*Enterococcus*	6	1189	3	4
*Listeria*	4	1195	0	3
*Erwinia*	5	1192	1	4
*Campylobacter*	8	1180	7	7
*Salmonella*	20	1147	13	22
*Lactobacillus*	5	1185	6	6
*Clostridioides*	2	1197	0	3
*Clostridium*	2	1195	2	3
*Yersinia*	2	1192	5	3
*Vibrio*	13	1163	6	20
*Rhizobium*	1	1198	1	2
*Klebsiella*	5	1176	8	13
*Xanthomonas*	1	1194	3	4
*Cronobacter*	1	1193	4	4
*Pectobacterium*	1	1194	2	5
*Pseudoalteromonas*	1	1192	4	5
*Brucella*	1	1194	1	6
*Ralstonia*	1	1193	2	6
*Cellulophaga*	1	1193	2	6
*Burkholderia*	1	1191	4	6
*Shigella*	1	1184	7	10
*Stenotrophomonas*	0	1199	0	3
*Citrobacter*	0	1196	1	5
*Mycobacterium*	0	1196	4	2

**Table 2 microorganisms-11-01398-t002:** Host prediction of newly isolated and sequenced phages.

Bacteriophage	Closest Relative	Real Host	PHERI Prediction
Dev-CS701	vB_CsaM_leB (KX431559.1)	*Cronobacter*	*Citrobacter*
vB_EcoM_VP1	vB_EcoM_JS09 (KF582788)	*E. coli*	*Escherichia*
vB-EcoM_KMB43	Rb49-like virus (AY343333)	*E. coli*	*Escherichia*
vB_KpnP_VP3	KPV811 (KY000081)	*Klebsiella*	*Klebsiella*
vB_EcoP_VP5	64795_ec1 (KU927499)	*E. coli*	*Escherichia*
PetSE1	vB_SenS-Ent1 (NC_019539.1)	*Salmonella*	*Salmonella*

## Data Availability

The tool PHERI is available at https://github.com/fmfi-compbio/pheri (accessed on 10 May 2023). The source code for the model training is available at https://github.com/andynet/pheri_preprocessing, and the source code for the tool is available at https://github.com/andynet/pheri (both accessed on 10 May 2023).

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
