# Peer review of "PHERI—Phage Host ExploRation Pipeline"

_microorganisms, 2023, doi:10.3390/microorganisms11061398_

Round 1

Reviewer 1 Report

The manuscript presents a new method for phage-host prediction (PHERI; what the acronym stands for?) based on machine learning methods. It is assumed that similar sequences in phages infecting the same host should be related to host specificity. The algorithm seems to have several limitations. One of them is that in the initial sample of phages from GenBank, there is an ample number of similar protein clusters that could lead to false predictions. Then, employing a decision tree to retain the essential predictors lets out considerable uninterpretable information. Considering only the set of 1,202 phages and host pairs, the algorithm had variable accuracy directly correlated with the number of sequences of the phage and their host. It is known that phage sequences are biased in databases towards those infecting pathogenic or industrially important bacteria, then the algorithm carries this limitation. Although the algorithm performs well in predicting hosts in a small sample of phages tested experimentally, it could be tested in metagenomic data. The authors emphasize the need to develop methods in this area, but they do not offer data to analyze the contribution of PHERI in complex data. The work would interest to readers in the area and be a starting point for future improvements.

Author Response

Dear Reviewer,

We would like to thank you for the review as well as for the time spent preparing the review in order to improve the submitted manuscript. In the document below, I tried to respond to your comments, and I firmly believe that the modification will be sufficient.

“The manuscript presents a new method for phage-host prediction (PHERI; what the acronym stands for?) based on machine learning methods.”

Response: This is no longer clear because the title lacks capitalization. It should be PHERI - Phage Host ExploRatIon pipeline. We corrected the title to make it clear what the PHERI acronym is.

“The authors emphasize the need to develop methods in this area, but they do not offer data to analyze the contribution of PHERI in complex data. The work would interest to readers in the area and be a starting point for future improvements.”

Response: The mentioned data is already available in the docker package as well as in the conda package.

https://hub.docker.com/r/andynet/pheri

https://anaconda.org/andynet/pheri).

Reviewer 2 Report

The manuscript describes a computer application called PHERI trained on a collection of phage sequences of known host, and intended to identify the host genus of metagenomic phage sequences. This is justified with a story about how phages are going to replace antibiotics and metagenomic phage sequences would be useful to help with that. The phage therapy connection is not terribly plausible. The authors cite two methods others have developed that can associate a virus sequence with a host, but neither of those produces an actual virus. And while it is all the rage to use metagenomics to characterize uncultivable viruses in uncultivable bacteria, the idea of producing an uncultivable virus for any purpose seems to be an oxymoron. And bacterial pathogens tend to be too cultivable, not uncultivable. While I have no objection to authors speculating about such things, this emphasis overtakes the introduction and the discussion, displacing relevant information about the more general bioinformatics problem, how their tool works, and how their tool would interact with the problem.

That being said, it is a point of interest to track metagenomic phages to host bacteria. And it is a theoretical point of interest to measure to what extent the genes in a phage with a known host cluster with other phages in that host as opposed to having been imported from elsewhere. The result of main interest to me is in fig. 4., which documents accuracy of assigning host genus to a test set of phages with known host specificity. PHERI seems to perform quite well.  It is presented in comparison to HostPhinder. HostPhinder claims to be better than using blast. Having spent considerable effort doing this without anyone's pipeline, I can tell you that the main sources of confusion about whether the similarity matches of a phage gene cluster with other phages in the same host are that matches can be found in host taxons related at the family or greater level because the phage gene was present in the ancestor of that higher taxon and descended vertically into multiple genera, and matching phage genes from host taxons at any distance are at some rate introduced by horizontal transfer. HostPhinder uses a very weak nucleotide comparison that can only see very close homologs, hence it is less likely to see these distractions. But the cost is that it won't find any signal for phages that don't have very close relatives in the training data. Instead, it will either find matches supplied by horizontal transfer and hence a source of misinformation, or nothing at all.

So the main technical feature to understand about the PHERI system is how it sets the threshold of detection of similarity. That is unfortunately opaque from the manuscript, although I suspect they did something fairly sophisticated. The manuscript describes both nucleotide and protein based comparisons to identify clusters of genes in the training sets, and then just says blast for comparing a test sequence to these training clusters. So the critical decisions of whether to use blastn or blastp and what thresholds for acceptance are used are not described. If I understand the comparison in fig. 4, they have excluded test sequences from any genus not in the training set (limited to bacterial genera with lots of phages sequenced), so they are ignoring the above mentioned tradeoff for viruses not hosted among the training genera.

The comparison engine is a decision tree described in the legends to a couple of figures in the supplemental data. They describe accepting a match based on a gini score for each gene cluster. That implies that they have made a distribution of the matching scores of the target phage to each of the genes in the training set, and compared it to a distribution somehow derived solely from the training set. I couldn't figure out how the latter operation was done. But perhaps the answer to the threshold question is that it is somehow auto determined from the distribution of similarity scores within the training set. Also, how they decided these particular clusters tested in this specific order is not obvious.

The system does appear to have merits. There is a github and a docker site to support users that want to download it. Source is available. There is not a server at an internet site. I didn't get deep enough to see if there was a good user manual, or if there was better documentation as to how exactly it works embedded somewhere in these sites. If there is, the authors ought to mention that.

Specific issues:

The manuscript would be a lot more readable if it had paragraphs with introductory and concluding sentences.

Can you explain better the criteria to put sequences in the same training cluster. For example, how many different portal clusters are there in Escherichia? Do they correspond to the major groupings: T7, P22, P2, T4, Mu, lambda, etc, or are they more finely divided, or all jumbled together into one cluster?

Fig. 1. I think they have confused Mycobacterium with Mycolicibacterium. Mycobacterium is a genus with a few thousand characterized phages. Mycolicibacterium is a related genus which, as far as I can tell, has no characterized phages.

In fig. 2, I don't understand why are there multiple shades of green points plotted, for example, but only one shade of green in the key.

The text is sloppy with the terms genus, species, and family.

line 111: "The phages with hosts from the 50 most abundant bacteria species were selected for further analysis. Does this mean phages from the 50 host genera most heavily populated with characterized phages were selected for further analysis?

line 171: "A novel phage genome sequence is classified using similar steps. At first, gene sequences are identified using Prokka tool. Then, collected gene sequences are compared with the gene clusters using Blast." This should say Prodigal, not Prokka; right? Prokka is just some value added, but not essential, as I understand it.

line 194: "Single-species phages were isolated by.." I think just means "Phages were purified by ..."

line 214: "In this study, we developed and benchmarked PHERI, a tool for predicting the bacterial host species of phages." You've only attempted to predict the host genus.

line 221: "Subsequently, host-specific clusters and cluster combinations were searched and created 50 family-specific decision trees" Those appear to be genus-specific, not family-specific.

legends to S2 and S3.

Why are some boxes greyed in?

Where on the figure does the result appear?

What are these Values = [x,y] entries?

Author Response

Dear Reviewer,

We would like to thank you for the review as well as for the time spent preparing the review in order to improve the submitted manuscript. In the document below, I tried to respond to your comments and I firmly believe that the modification will be sufficient.

„The manuscript describes a computer application called PHERI trained on a collection of phage sequences of known host, and intended to identify the host genus of metagenomic phage sequences. This is justified with a story about how phages are going to replace antibiotics and metagenomic phage sequences would be useful to help with that. The phage therapy connection is not terribly plausible. The authors cite two methods others have developed that can associate a virus sequence with a host, but neither of those produces an actual virus. And while it is all the rage to use metagenomics to characterize uncultivable viruses in uncultivable bacteria, the idea of producing an uncultivable virus for any purpose seems to be an oxymoron. And bacterial pathogens tend to be too cultivable, not uncultivable. While I have no objection to authors speculating about such things, this emphasis overtakes the introduction and the discussion, displacing relevant information about the more general bioinformatics problem, how their tool works, and how their tool would interact with the problem.“

Response: The main task of the PHERI tool was not to find bacteriophages in the metagenome infecting uncultivable bacteria. Rather, it is a tool to accelerate the isolation of specific bacteriophages from mixed samples (for example, concentrated wastewater). Pheri is supposed to select bacterial genera suitable for capturing phages present in the sample. In this way, we would avoid the trial-and-error method in the purification of phages from the environment. Moreover, we did not try to claim, nor do we share the opinion, that bacteriophages will replace antibiotics. However, they represent one of the possible solutions to the therapy of multiresistant bacteria (as we indicated in L35-36)

„So the main technical feature to understand about the PHERI system is how it sets the threshold of detection of similarity. That is unfortunately opaque from the manuscript, although I suspect they did something fairly sophisticated.“

Response: How PHERI determines the threshold of detection and similarity is described on lines 141-147 (In the first step, genes were deduplicated using CD-hit to reduce the enormous number of sequences. Then, gene pairs with at least some local sequence similarity were identified using an optimized implementation of the Blast alignment tool, called Croco-BLAST. Afterwards, the identified pairs with high local sequence similarity under-went thorough pairwise alignment to retrieve more accurate similarity scores. Based on these similarity scores, we identified gene clusters using the Markov Cluster Algorithm implemented in package MCL). The most important part is the MCL package, which receives a similarity score at the input from the global alignment of nearby proteins and then sets a "threshold" for putting proteins into clusters. MCL has an inflation parameter that determines the granularity of the clusters. A higher value gives fewer clusters. In our case, the value 1.2 was used

„The manuscript describes both nucleotide and protein based comparisons to identify clusters of genes in the training sets, and then just says blast for comparing a test sequence to these training clusters. So the critical decisions of whether to use blastn or blastp and what thresholds for acceptance are used are not described.“

Response: The thresholds for acceptance  was blastp -max_target_seqs 1000000 -max_hsps 1 . All other parameters are default. We added the setting to the methods of the article (L-144-145)

They describe accepting a match based on a gini score for each gene cluster. That implies that they have made a distribution of the matching scores of the target phage to each of the genes in the training set, and compared it to a distribution somehow derived solely from the training set. I couldn't figure out how the latter operation was done. But perhaps the answer to the threshold question is that it is somehow auto determined from the distribution of similarity scores within the training set. Also, how they decided these particular clusters tested in this specific order is not obvious.“

Response: This is a standard decision tree algorithm that automatically chooses which cluster is important based on statistics. The tool chooses a phage representation, for example (phage i is an array of numbers a[i,j]) and a label (phage i is an ecoli phage) and learns the relationship between the two inputs.

„The system does appear to have merits. There is a github and a docker site to support users who want to download it. Source is available. There is not a server at an internet site. I didn't get deep enough to see if there was a good user manual, or if there was better documentation as to how exactly it works embedded somewhere in these sites. If there is, the authors ought to mention that.“

Response: A new link https://github.com/fmfi-compbio/pheri will be listed in the methods. Test data as well as a brief manual will be part of the package

„Can you explain better the criteria to put sequences in the same training cluster. For example, how many different portal clusters are there in Escherichia? Do they correspond to the major groupings: T7, P22, P2, T4, Mu, lambda, etc, or are they more finely divided, or all jumbled together into one cluster?“

Response: Bacteriophages were added to the training set randomly using the random split method. We chose a random selection to avoid possible unwanted influence on the results by the selection of bacteriophages.

„Fig. 1. I think they have confused Mycobacterium with Mycolicibacterium. Mycobacterium is a genus with a few thousand characterized phages. Mycolicibacterium is a related genus which, as far as I can tell, has no characterized phages.“

Response: The reviewer is right, Mycobacterium was mistaken for Mycocilibacterium. We apologize, and we replaced figure 1 with the correct one in the manuscript.

„In fig. 2, I don't understand why are there multiple shades of green points plotted, for example, but only one shade of green in the key.“

Response: The multiple shaded green points still represent the host of bacteriophages from the Mycobacterium genus, but due to their overlap, they have a translucency. It is thus possible to see when they overlap.

„line 111: "The phages with hosts from the 50 most abundant bacteria species were selected for further analysis. Does this mean phages from the 50 host genera most heavily populated with characterized phages were selected for further analysis?“

Response: Yes, it means that phages from the 50 most abundant host genera with characterized phages were selected for further analysis. We also replaced Species with Genus in the manuscript, according to the reviewer's proposal

„line 171: "A novel phage genome sequence is classified using similar steps. At first, gene sequences are identified using Prokka tool. Then, collected gene sequences are compared with the gene clusters using Blast." This should say Prodigal, not Prokka; right? Prokka is just some value added, but not essential, as I understand it.“

Response: The sentence was corrected to the Prodigal tool as suggested by the opponent

„line 194: "Single-species phages were isolated by.." I think just means "Phages were purified by ..."

Response: We changed the text according to the opponent's proposal

„line 214: "In this study, we developed and benchmarked PHERI, a tool for predicting the bacterial host species of phages." You've only attempted to predict the host genus.“

Response: The sentence was changed to:  In this study, we developed and benchmarked PHERI, a tool for predicting the bacterial host genera for phages from metagenomic data, according to reviewer´s proposal

„line 221: "Subsequently, host-specific clusters and cluster combinations were searched and created 50 family-specific decision trees" Those appear to be genus-specific, not family-specific.“

Response: We changed the text according to the opponent's proposal

„legends to S2 and S3. Why are some boxes greyed in? Where on the figure does the result appear?

What are these Values = [x,y] entries?“

Response: The trees represents the resulting classifier for a single host genus. Each node compares a number of genes of a phage in a particular cluster and a learned threshold. If the number of genes in cluster is smaller than the threshold, the classification decision moves to the left, otherwise to the right. Upon reaching a leaf, the final decision if the phage's host belongs to the genus or not is made. The values [x, y] represent the number of phages from training set for each case at a particular node. For example, at the tree root we can see that 4706 phages from training set did not infect this particular host and 17 phages did. Gray boxes represent those phage protein sequences whose annotated function is known

Reviewer 3 Report

In their manuscript Baláž et al. developed a sequence-based machine-learning tool for prediction of bacterial host species of phages. Such in silico approaches are very important for the advancement of tailored and efficient uses of phages in fields such as phage therapy or others. The work represents a very valuable starting point for such predictions, however a shortcoming of the paper is the limited outlook on further development of the tool e.g. in respect of further refinement of predictions down to the bacterial host strain level, adverse predictions from bacterial genome to phage families or even prediction of receptor binding. The outlook could also refer to developments how prediction tools could guide engineering approaches of phages. Addition of views by the authors in this regard would improve the current paper, otherwise the reviewer highly recommends this work for publication. As a minor comment, weblinks in the abstract should be moved to the method section.

Author Response

Dear Reviewer,

We would like to thank you for the review as well as for the time spent preparing the review in order to improve the submitted manuscript. In the document below, I tried to respond to your comments, and I firmly believe that the modification will be sufficient.

“The work represents a very valuable starting point for such predictions, however a shortcoming of the paper is the limited outlook on further development of the tool e.g. in respect of further refinement of predictions down to the bacterial host strain level, adverse predictions from bacterial genome to phage families or even prediction of receptor binding. The outlook could also refer to developments how prediction tools could guide engineering approaches of phages. Addition of views by the authors in this regard would improve the current paper.”

Response: We discussed the possibilities of using the tool and its further development in the last sentences of the discussions. Nevertheless, we consider it appropriate to supplement the text:

“There is also the potential to refine the prediction with the increasing number of new phages in databases. With the help of more accurate data from the database and the addition of several bacterial genera and species, we plan to increase the accuracy of the tool to the level of bacterial species. One of the advantages of host prediction based on the clustering of individual genes is the possibility of highlighting genes with unknown functions necessary for infection. The identification of such genes may, in the future, help scientists elucidate the mechanisms of infection by individual bacteriophages. Moreover, by identifying these genes, it will be possible to study them better or directly use them for the preparation of recombinant phages with changed properties.”

“As a minor comment, weblinks in the abstract should be moved to the method section.”

Response: The sentence: “The tool PHERI, the source code for the model training as well as  the source code for the tool is available at https://github.com/fmfi-compbio/pheri “was added into method section (L168-169)

Reviewer 4 Report

THE manuscript titled PHERI - Phage Host Exploration pipeline refers development of a software tool for biological study/application. From my perspective, this study is important and useful. The author correlated the prediction results with experimental data which suggests that the tool is very accurate. I have no major comments on the manuscript. Some tinny editorial corrections should be made.

Author Response

Dear Reviewer,

We would like to thank you for the review as well as for the time spent preparing the review in order to improve the submitted manuscript. In the document below, I tried to respond to your comments, and I firmly believe that the modification will be sufficient.

THE manuscript titled PHERI - Phage Host Exploration pipeline refers development of a software tool for biological study/application. From my perspective, this study is important and useful. The author correlated the prediction results with experimental data which suggests that the tool is very accurate. I have no major comments on the manuscript.

„Some tinny editorial corrections should be made.“

Response: We made several editorial corrections based on the recommendation of other reviewers. (for example in lines 22-25, web link was changed and moved into methods, L112 word species was changed to genus, L119 Fig1. was corrected....)